# Respiratory Health in a Community Living in Close Proximity to Gold Mine Waste Dumps, Johannesburg, South Africa

**DOI:** 10.3390/ijerph17072240

**Published:** 2020-03-26

**Authors:** Samantha Iyaloo, Tahira Kootbodien, Nisha Naicker, Spo Kgalamono, Kerry S. Wilson, David Rees

**Affiliations:** 1Occupational Medicine Section, National Institute for Occupational Health, National Health Laboratory Service, Johannesburg 2001, Gauteng Province, South Africa; spok@nioh.ac.za (S.K.); davidr@nioh.ac.za (D.R.); 2School of Public Health, Faculty of Health Sciences, University of the Witwatersrand, Johannesburg 2193, Gauteng Province, South Africa; NishaN@nioh.ac.za (N.N.); kerryw@nioh.ac.za (K.S.W.); 3Epidemiology and Surveillance Section, National Institute for Occupational Health, National Health Laboratory Service, Johannesburg 2001, Gauteng Province, South Africa; tahirak@nioh.ac.za; 4Environmental Health Department, Faculty of Health Sciences, University of Johannesburg, Johannesburg 2028, Gauteng Province, South Africa

**Keywords:** gold mine waste dump dust, chronic respiratory conditions, cumulative exposure index, crystalline silica dust

## Abstract

The effects on respiratory health in populations living close to silica-rich gold mine dumps are unknown. This pilot study related respiratory health and exposure to mine dump dust using two measures of exposure: exposure group, based on distance lived from the mine dump—high (*n* = 93) (home <500 m from a mine dump), moderate (*n* = 133) (500–1.5 km), and low (*n* = 84) (>15 km, control group); and cumulative exposure index (CEI) derived from exposure group and number of years of residence in each exposure group. Participants were interviewed about respiratory symptoms and had chest X-rays and spirometry. We adjusted for key respiratory confounders. No subject had radiological features of silicosis. The high relative to low exposure group had significantly elevated adjusted odds ratios (aORs) for upper respiratory symptoms (aOR: 2.76, 95% CI: 1.28–5.97), chest wheezing (aOR: 3.78; 95% CI: 1.60–8.96), and spirometry-diagnosed chronic obstructive pulmonary disease (COPD) (aOR: 8.17; 95%CI: 1.01–65.85). These findings were similar for the high relative to medium exposure group, but no significant associations were found for the medium versus low exposure group. Chronic bronchitis and tuberculosis risks did not differ significantly among groups. CEI and exposure group produced similar results. In conclusion, residents residing <500 m from mine dumps had elevated adverse respiratory health effects.

## 1. Introduction

In South Africa, mining processes from the Witwatersrand gold basin have resulted in approximately 270 tailings storage facilities, more commonly referred to as gold mine waste dumps, or colloquially, mine dumps, around Johannesburg [1,2]. Gold mine waste dump dust is fine, dry, tiny particles of earth matter that have been processed and is prone to be carried in the air. Gold mine waste dump dust, specifically around the Witwatersrand area has a high crystalline silica content [3,4,5]. These mine waste dumps are mostly unlined and most are not vegetated, thus being a significant source of silica-rich, air-borne dust [2,3,6,7] consisting of fine particulate matter [7,8]. The reworking of gold mine dumps for further gold extraction using newer technologies has also resulted in an increase in the proportion of inhalable particles in the tailings material [8].

Residential encroachment in the areas surrounding mine dumps over a 50-year period may pose a great health risk for the individuals who live in these areas [8].

Research in the Witwatersrand area has shown that environmental standards for inhalable particulate matter 10 µm or less (PM_10_) have at times been exceeded in the vicinity of these gold mine waste dumps [6,7,8]. Inhalable particulate matter may travel up to 2 km downwind of mine waste dumps; up to half a million people who live around these waste dumps may be affected [8]. The distance the dust may travel is, however, weather-dependent. Dust generated from mining does not negatively affect ambient air quality during normal Nagpur Tropical weather conditions, collected by the Indian Meteorological Department, in areas more than 500 m from the mine [9]. Ojelede et al. [6] found that South African Department of Environmental Affairs 24-hour threshold value for airborne aerosols of 120 μg/m^3^ was not exceeded except where wind speeds reach 7 m/s. However, exceedances reached 2160 μg/m^3^ (on one single occasion during a 5-day measurement period), which is 18 times the threshold value.

There is no limit value for non-occupational silica dust exposure in South Africa currently. However, an international interim limit of 0.12 μg/m^3^ has been set by the United States Agency for Natural Resources [10] and 5 μg/m^3^ by the United States Environmental Protection Agency [11]. Kneen et al. found that PM_10_ hourly average concentrations within 1.5 km of mine dumps in the Witwatersrand area, ranged between 5 and 377 μg/m^3^ during a four-week period of monitoring ambient air-borne particles [7]. Their modeled estimates determined that occupational health exposure standard thresholds for dust generation were likely to be exceeded for about 845 hours/year, which is equivalent to 106 8-hour shifts, in an area near Soweto, Johannesburg. Andraos et al. found that the 24-hour average ambient crystalline silica PM_10_, ranged from 4.63–19.4 μg/m^3^ in a study of gold mine dumps in Johannesburg [3].

Little is known about the respiratory health effects of exposure to the siliceous gold mine dump dust. In particular, no studies have examined respiratory diseases commonly associated with silica, such as silicosis, tuberculosis, and chronic obstructive pulmonary disease (COPD) [12,13], or decline in spirometric parameters, such as forced vital capacity (FVC) and forced expiratory volume in one second (FEV1). The studies that have been done have relied on symptom reporting; they have shown increased symptoms associated with proximity to gold mines or dumps. A Portuguese health survey of residents living near an abandoned gold mine found a higher prevalence of respiratory and irritant-induced symptoms compared to a community 45 km away from the gold mine [14]. Two South African studies examined the association of proximity to a gold mine waste dump and respiratory conditions. The first found a positive association between chest wheezing (OR: 1.38; 95% CI: 1.10, 1.71) and rhino-conjunctivitis (OR: 1.54; 95% CI: 1.29, 1.82) compared to a community 5 km away from a gold mine waste dump [15]. In the second study, Johannesburg residents living within 1–2 km of a gold mine dump were found to be at higher risk of chronic respiratory symptoms (chest wheezing and chronic cough), and diseases (chronic bronchitis and self-reported emphysema) [16]. The prevalence of chronic bronchitis in those living close to the gold mine dump was 13.4% compared to a prevalence of 7.5% in the population further away from the dump.

There is a paucity of research on the respiratory health effects on residents in close proximity to gold mine waste dumps using objective measures of health outcomes like spirometry and chest X-rays which motivated this study. This study aimed to examine the associations between respiratory conditions and residence near gold mine dumps. This study generated new information on the environmental risk of exposure from gold mine dump dust on respiratory health based on distance of exposure and a derived CEI.

## 2. Materials and Methods

### 2.1. Study Population and Setting

We conducted this cross-sectional household survey as part of larger panel study called the Johannesburg Health, Environment and Development (HEAD) study whose methodology and objectives are described elsewhere [17]. We did the study in the Riverlea and Ennerdale townships of Johannesburg, South Africa. Ennerdale was a new township added for the purposes of this sub-study and was not part of the original HEAD study. Riverlea is primarily an old apartheid-era “mixed race” (colored in South African parlance) community with a population of about 16,230 at the time of the study living in approximately 4208 households built around and in close proximity to established gold mines (which predate the community), which have since become mine waste dumps [18,19]. Ennerdale is 34 km from Riverlea and had a population of about 71,815 and approximately 19,844 households. We chose Ennerdale as a low-exposed population based on its large distance from a mine or mine dump (>15 km) and for its similar socio-demographic profile to the Riverlea population [20] (Figure 1). We selected households randomly using a randomly generated numbers in Microsoft Excel and City of Johannesburg aerial town planning maps.

### 2.2. Exposure Estimation

We estimated exposure using two methods. The first was based on household distance from the nearest gold mine waste dump. Residents living within 500 m of a mine waste dump boundary, we defined as our high exposure group. The rest of Riverlea lies more than 500 m, but within 1.5 km of the gold mine waste dump boundary, and we defined this group as the medium exposure group. The Ennerdale community was regarded as the low exposure group.

For the second exposure estimation method, we derived a cumulative exposure index (CEI) for each study participant using the product of the number of years lived in each exposure group and a dust intensity weighting for each of the three groups. We established the weighting by using the modeled dust levels for communities around gold mine dumps as determined by Ojelede et al. [20]. Ojelede used predictive models of PM_10_ concentrations in the area surrounding mine dumps in Riverlea and further away, using actual particulate matter concentrations collected closer to the mine dumps. Inhalable dust measurements collected from communities closest to a gold mine waste dump (in the high exposure group) had a predictive model value of 60 μg/m^3^. Those further away from the mine dump (in the medium exposure group) had a predictive model value of 40 μg/m^3^, and those outside the demarcated zone surrounding gold mine dumps (similar to our low-Ennerdale exposure group) had a predictive model value of 10 μg/m^3^ of PM_10_ dust. Based on the above, residents of Riverlea in the high and medium exposure groups were given an intensity weighting of six and four, respectively, and residents in Ennerdale (low exposure group) were given an intensity weighting of one. CEIs for study participants with one or more category of exposure (in cases where residents moved from one exposure area to another exposure area) were calculated using the following equation:(1)$$CEI=\left(residence\left(1\right)\;weighting\;x\;years\;\;of\;stay\left(1\right)\right)+\left(residence\left(i\right)\;weighting\;x\;years\;of\;stay\left(i\right)\right)$$

### 2.3. Data Collection

Following written informed consent, participants completed a questionnaire administered by a trained interviewer in their preferred language. The questionnaire covered demographic information, residential history, occupational and domestic exposures, smoking history, and the outcomes of interest, as detailed in Table 1. Participants who completed the interview were invited to a centrally located community hall for chest X-rays and spirometry to objectively identify features of chronic lung disease (particularly airflow limitation on spirometry and tuberculosis and silicosis on chest X-ray). We conducted the investigations over two weekends so that participants had alternatives dates to choose from. Posterio-anterior chest X-rays were taken using South African National Accreditation System (SANAS) approved service providers. Chest X-rays were read by two occupational medicine specialists and one radiologist who were all experienced and had received training in the International Labour Organization’s Classification of Radiographs of Pneumoconioses (ILO International Classification of Radiographs of Pneumoconioses 2011) [21]. Spirometry was done by experienced, accredited, and trained service providers according to the 2005 ATS/ERS standards [22] using a KoKo Legend II spirometer. Percentage predicted forced vital capacity (FVC) and forced expiratory volume in one second (FEV1) values were calculated using the Global Lung Function Initiative (GLI) 2012 reference equations [23]. If the FEV1/FVC ratio was <70% and/or FEV1 was <80% of predicted, a bronchodilator was administered. The lower limit of normal of the FVC, FEV1, and FEV1/FVC were calculated using the GLI 2012 reference equations.

### 2.4. Statistical Analysis

We conducted the analyses using Stata version 14 (Stata Corporation, College Park, Texas, USA). We used Chi-squared tests for trend across exposure groups. Associations between the respiratory outcomes of interest, exposure group, and CEI were estimated, while controlling for the a priori confounders of age; sex; a smoking history; an occupational history of exposure to vapors, gas, dust, or fumes; biomass fuels exposure; tuberculosis; and socio-economic status. We included a previous history of tuberculosis or evidence of radiological tuberculosis as a confounder for COPD since tuberculosis may appear similar to COPD on spirometry. We also conducted a sensitivity analysis excluding those with tuberculosis while assessing the effect of CEI on COPD. A *p* < 0.05 was used as the pre-determined significance level due to the small sample size. We used multiple linear regression to estimate the effect of CEI on the percentage predicted FEV1 and FVC. Modeling with robust standard errors was used to account for deviations in the assumptions. Interactions for smoking status on the relation between CEI and the percentage of predicted FEV1 and FVC were investigated while conducting the multiple linear regression.

### 2.5. Ethical Approval and Permissions

We obtained ethical approval from the University of the Witwatersrand’s Human Research Ethics Committee (medical) (M150550). We obtained written informed consent from each subject and referred all study participants with abnormalities (including abnormal spirometry detected during the examination), to the nearest healthcare facility for further investigation and management if required.

## 3. Results

Three hundred and eighty-one residents from separate households were asked to participate in the study; 55 were not available and 16 declined, resulting in 310 (81%) positive responses. Table 2 shows that of these 310 participants, 93 (30.0%), 133 (42.9%), and 84 (27.1%) were in the high, medium, and low exposure groups, respectively. All 310 study participants completed a questionnaire, but fewer had chest X-rays: 55 (59.1%) in the high; 69 (51.9%) in the medium; and 53 (63.1%) in the low exposure group. We found no significant differences in the symptoms (cough and shortness of breath) between those that came for the chest X-ray compared to those who did not come for the chest X-ray. A further 12 (21.8%), 10 (14.5%) and 6 (11.3%) study participants from the high, medium, and low exposure groups, respectively, did not do spirometry due to contra-indications, the most common of which were severe hypertension or suspect active tuberculosis.

Table 2 lists the socio-demographic, environmental, and occupational characteristics of the study participants by exposure group. In all the exposure groups, close to two-thirds of study participants were women. A smaller percentage of the low (46.4%) and medium (43.6%) exposure group smoked than their high exposure counterparts (64.5%). Other notable differences between the high compared to the medium and low exposure groups are tertiary education (5.5% vs 27.8% and 21.4%), lower percentage of highest category monthly household income (4.4% vs 24.0% and 19.0%), and longer duration of residence in the study group (38 years vs 20 and 16 years). Biomass fuel use was lowest in the high exposure group (4.9%), but this group had by far the highest percentages with occupational exposure (high 53.9% vs low 17.9%). The high exposure group has a high percentage (71.4%) of houses with asbestos roofs. These roofs are, however, well maintained and are regularly painted by the government and, therefore, do not pose a hazard to the people living in the house.

The number and percentage of individuals with the respiratory symptoms and conditions of interest are captured in Table 3. The percentages of individuals with the health outcomes did not differ significantly across exposure groups on chi-square test for trend, except for chronic bronchitis which was most prevalent (20.4%) in the high exposure group (x^2^ = 8.06, *p* = 0.018). The prevalence of probable past or current radiologic tuberculosis was 10.9%, 17.4% and 7.5% in the high, medium and low exposure groups, respectively (x^2^ = 2.84, *p* = 0.24). COPD diagnosed on spirometry was more common in the high exposure group (18.6%) than in the in the medium (10.2%) and low (10.6%) exposure groups, but not significantly so (x^2^ = 1.87, *p* = 0.39). The median percentage predicted FEV1 and FVC were highest in the high exposure group (98.0% and 102.0%, respectively). We used crude odds ratios to examine the effect of exposure category on 11 health outcomes. The effect estimates and their 95% confidence intervals are presented in Table 3. There were significant differences in the high versus low exposure groups for upper respiratory tract symptoms (OR: 2.83; 95% CI: 1.58, 5.08), ocular symptoms (OR: 2.63; 95% CI: 1.52, 4.58) and a wheezy chest (OR: 3.21; 95% CI: 1.62, 6.34). Significant associations in the high vs medium exposure groups were upper respiratory tract symptoms (OR: 2.63; 95% CI: 1.52, 4.58), chronic cough (OR: 2.80; 95% CI: 1.49, 5.27), chronic bronchitis (OR: 3.16; 95% CI: 1.39, 7.16), and a wheezy chest (OR: 2.83; 95% CI: 1.58, 5.08). There were no significant differences when comparing the medium and low exposure groups.

We did multivariate logistic regression, shown in Table 4, which revealed significant differences between the high and low exposure groups in the adjusted risk of upper respiratory tract symptoms (aOR: 2.76; 95% CI: 1.28, 5.97), ocular symptoms (aOR: 4.68; 95% CI: 1.87, 11.068), wheezy chest (aOR: 3.78; 95% CI: 1.60, 8.96), and COPD diagnosed on spirometry (aOR: 8.17; 95% CI: 1.01, 65.85). We conducted a sensitivity test, excluding those with COPD who either had a previous history of tuberculosis or those with radiological evidence of tuberculosis. This analysis demonstrated the same trend (aOR: 8.83; 95% CI: 0.70–112.08). Significantly elevated aORs for the high relative to the medium exposure group were estimated for upper respiratory tract symptoms (aOR: 2.12; 95% CI: 1.01, 4.42), ocular symptoms (aOR: 3.01; 95% CI: 1.32, 6.89), and wheezy chest (aOR: 3.60; 95% CI: 1.60, 8.11). We found a negative association for chronic cough and being part of the medium relative to the low exposure group (aOR: 0.36; 95% CI: 0.17, 0.77).

We calculated crude and adjusted odds ratios for respiratory health outcomes using cumulative exposure index (CEI) as the exposure variable. The median of the CEI was 80 exposure group-years (IQR: 4–408) and was 228 (IQR: 90–408), 80 (IQR: 12–260) and 16 (IQR: 4–44) exposure group-years in the high, medium, and low exposure groups, respectively. Results of these analyses are in Table 5. CEI was associated with elevated risks of upper respiratory tract symptoms (aOR: 1.0034; 95% CI: 1.00051, 1.0064), wheezy chest (aOR: 1.0043; 95% CI: 1.0011, 1.0075), and COPD diagnosed on spirometry (aOR: 1.010; 95% CI: 1.0014, 1.019). The sensitivity test excluding those with COPD who either had a previous history of tuberculosis or those with radiological evidence of tuberculosis was also significantly associated with CEI (aOR: 1.011; 95% CI: 1.000032, 1.023).

We examined the potential negative effect on FEV1 and FVC using exposure groups and CEI as the exposure rubrics and percentage predicted spirometry values as the outcomes using multiple linear regression. We controlled for monthly income, pack years of smoking, use of biomass fuels in the house, and an occupational exposure history to vapors, gas, dust, and fumes. Table 6 shows that CEI was not significantly associated with FEV1 percent predicted (*p* = 0.27). Counter-intuitively, a positive significant relation was found between CEI and FVC percent predicted: there was a 0.062% (*p* = 0.008) increase in the percentage of predicted FVC for every one-unit increase in CEI (after adjusting for potential confounders).

## 4. Discussion

This pilot cross-sectional study of a community exposed to gold mine waste dump dust is the first to our knowledge to include chest radiographs and spirometry in the evaluation of respiratory health. The residents of Riverlea living closest to the mine dump (high exposure group) had higher risks of upper respiratory and ocular symptoms, a wheezy chest, and, importantly, COPD diagnosed on spirometry. Multiple linear regression did not reveal exposure-associated lower percent predicted FVC and FEV1. Our derived CEI displayed similar results to the crude exposure estimate of (close) distance from the mine dump since it was a risk factor in upper respiratory symptoms, wheezy chest, and, again, COPD diagnosed on spirometry.

### 4.1. Symptoms

Our study found elevated risks of upper respiratory tract symptoms and wheezy chest associated with mine dust exposure, measured as either exposure group or CEI. The results of our study are in keeping with other studies where an association between respiratory symptoms (high levels of current wheeze and rhino-conjunctivitis) [15,16] have been identified in communities living in close proximity to gold mine dumps in South Africa. There were, however, no significantly increased ORs for symptoms in the medium relative to the low exposure group except for the spurious finding of the medium exposure group having lower odds than the low exposure group of having chronic cough. These results tentatively suggest that only the residents living within 500 m of the gold mine waste dump may be at increased risk of these respiratory symptoms.

We did not find exposure-associated significantly elevated risks of chronic cough or chronic bronchitis in multivariate analyses, although the adjusted effect estimates were >1 for chronic bronchitis in the high relative to the low exposure group and in the CEI regression model. The lack of significant associations is contrary to that reported by Nkosi et al. [16]. They interviewed 2397 adults ≥55 years of age and found elevated adjusted ORs for those within 1–2 km of mine dumps (exposed) compared to those 5 km away (unexposed) for chronic cough (aOR: 2.02; 95 % CI: 1.58, 2.57) and chronic bronchitis (aOR: 1.74; 95% CI: 1.26, 2.39). However, their study population were older adults who had a higher number of exposure years. Despite the lack of positive associations in our study, the subjects from Riverlea had high prevalences of the two conditions: 23.5% for chronic cough; and 12.9% for chronic bronchitis, rising to 20.4% in the high exposure group, which exceeded the 14.3% in Ennerdale. These prevalences are similar to those reported by Nkosi et al. [15] at 26.6% and 13.4% in their exposed subjects but much higher for chronic bronchitis (2.3% for men and 2.8% for women) identified in a general adult South African population survey published in 2004 by Ehrlich et al. [23]. The unexpectedly high prevalences of chronic cough (32.1%) and chronic bronchitis (14.3%) in Ennerdale may have obscured possible dust effects and are difficult to explain. There were 32.1% of Ennerdale residents who were current smokers compared to 33.1% in the Riverlea medium exposure group, and the area has no nearby sources of industrial pollution. The elevated, albeit not significantly so, ORs and the high prevalences of chronic cough and bronchitis means that we cannot exclude an effect of the mine dump dust on these conditions.

### 4.2. Silicosis

We found no cases of radiological silicosis in the Riverlea and Ennerdale communities. This finding may be due to our small sample size, although there is no published literature on silicosis induced by living near gold mine dumps. Environmental silicosis does occur, however, following prolonged inhalation of desert sand dust [23,24]. According to a review on non-occupational silicosis, cases of silicosis have also been documented in communities close to slate pencil and agate industries [25,26]. Identifying cases of silicosis in Riverlea would, therefore, not have been very surprising. One explanation for the absence of cases in the Riverlea community is that not enough of them had had sufficiently long exposure to develop the disease. This explanation is unlikely though, as 90% of the high exposure group had >20 years’ residence and 57% of the medium exposure group.

### 4.3. Tuberculosis

Radiologically identified pulmonary tuberculosis prevalence did not increase consistently with exposure group, being highest in the medium group at 17.4%. This prevalence of pulmonary tuberculosis is higher than the prevalence of radiologically detected (old and new) tuberculosis (12.9%) in a low socioeconomic community in Cape Town, South Africa, which had the highest tuberculosis notification rate [27]. Self-reported tuberculosis was highest in Ennerdale: 7.1% versus 3.5% for Riverlea. In addition, we found no significant associations between the exposure variables and radiological and self-reported healthcare practitioner diagnosed tuberculosis in multivariate analyses. Our methods of ascertaining tuberculosis have limitations: some misclassification of disease is likely using radiologic features; and unreliable recall may occur on questioning.

### 4.4. COPD

Our study identified a COPD prevalence of 18.6%, 10.2%, and 10.4% in the high, medium, and low exposure groups, respectively, a combined prevalence for Riverlea of 13.7%. These values are lower than those of the Burden of Obstructive Lung Disease (BOLD) population-based survey done in Cape Town, South Africa, which found a prevalence of 23.8% [28]. Of relevance is the high percentages of current smokers in the Cape Town survey: 56.9% in men; and 40.6% in women. We observed exposure-associated elevated risks for COPD in multivariate analyses adjusted for potential confounders using either exposure group or CEI as the exposure variable. The aOR for high vs low exposure groups was quite large at 8.17 (95% CI 1.01, 65.85). A bigger study, preferably with more refined dust concentration data is needed to confirm the association between mine dump dust and COPD.

### 4.5. Spirometry

Multivariate regression analyses showed no effect of CEI on %FEV1, but a counter-intuitive positive effect on percentage of predicted FVC. We consider the latter effect to be spurious. We found no literature on environmental dust exposure from mines or their waste dumps and effects on spirometric parameters of surrounding communities. However, the effect of ambient particulate matter more generally on lung function in adults has been investigated, producing inconsistent findings [29]. Nevertheless, positive studies, particularly from Asia, have been reported recently showing a decline in FVC and FEV1 [29,30] and an increase in COPD associated with elevated PM_10_ and PM_2.5_ concentrations [31].

Some limitations should be borne in mind when considering the results of this study. We did not have objective measures of exposure and the allocation to high and medium exposure groups may have been imprecise. Misclassification of exposure, therefore, possibly occurred for the Riverlea groups. This misclassification would obscure differences between the high and medium groups, assuming some medium exposed individuals should have been in the high exposure group. Recent objective measures of exposure could be misleading in any event as many of the participants had decades of residence in Riverlea, and dust levels probably varied over time quite substantially. Our exposure group classification based on distance from the mine dump has been used before in other similar studies [15,16]. The other measure of exposure, the CEI, has not been used before, but it has the potential to provide a more reliable estimate of exposure since it is an amalgam of duration and intensity of exposure. Both these exposure estimates remain to be validated.

The somewhat disappointing uptake of spirometry and chest radiograph of about 50% may have meant that participants who had these investigations were not representative of those interviewed. The potential effect of this is unclear; but possibly subjects with symptoms preferentially had the investigations resulting in artificially high prevalences of spirometric abnormalities. This is unlikely, though, because there were no significant differences in the prevalences of chronic cough and dyspnoea among the respondents (participants who conducted questionnaires and who came for the CXRs and LFTs) and non-respondents (participants who conducted questionnaires but did not come for the CXRs and LFTs).

The use of the lower limit of normal of FEV1/FVC to define COPD is a more accurate measure of the disease than the GOLD criterion of < 70%, since it takes account of an age-related decline in FEV1/FVC below 70%, thus reducing over-estimates of COPD in older people [32,33]. It does mean, though, that our COPD rates may not be comparable to those determined by the GOLD criterion.

## 5. Conclusions

This study showed that respiratory and ocular symptoms and objective measures of respiratory disease were higher in the most exposed group of study participants. Objective measurements of inhalable and respirable dust and respirable crystalline silica are required to refine the use of distance of residence from mine dumps as an exposure tool and to improve the exposure weighting for calculating the CEI. Larger studies are required to explore the COPD mine dump dust exposure association found in this study.

## Figures and Tables

**Figure 1 ijerph-17-02240-f001:**
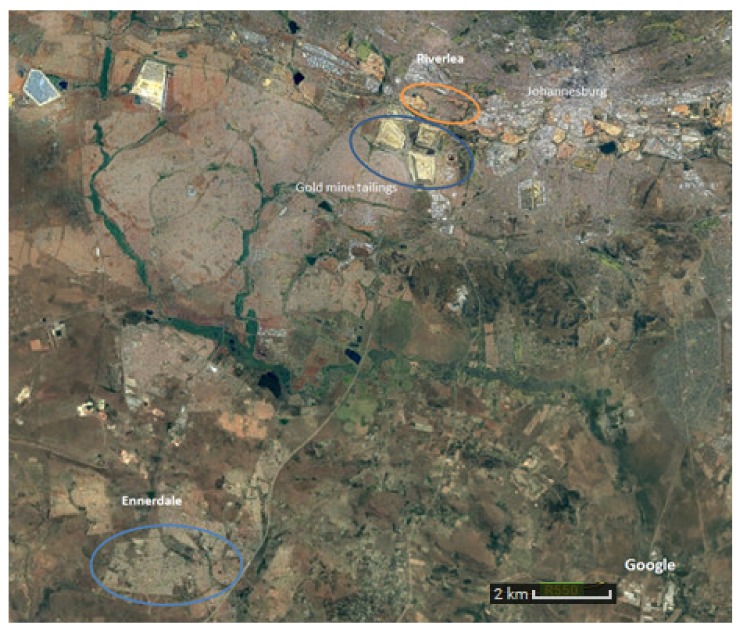
Map of the study area indicating the study sites and gold mine dumps (source: Google Earth v 9.2.65.2, Johannesburg, Gauteng Province, South Africa. 26°08’45”S 28°16’08”E, Eye altitude 1662 m. Digital Globe 2012. http://www.earth.google.com (August 20, 2018)).

**Table 1 ijerph-17-02240-t001:** The definitions and data collection methods for the outcomes of interest.

Variables	Data Collection Method	Definition
Independent Variables		
Cumulative exposure index	Study questionnaire	Derived using the quotient of number of years lived in an exposure group and the intensity weighting as determined by Ojelede et al. [6]
Occupational history of exposure to vapours, gas, dust and fumes	Study questionnaire	Exposure for a year or more to vapors, gas, dust, and/or fumes.
Exposure to biomass fuels	Study questionnaire	If a biomass fuel is used for either cooking or heating.
Mean monthly household income	Study questionnaire	We used mean monthly household income as a proxy for socio-economic status.
Smoking history	Study questionnaire	Positive history is at least one pack-year of smoking cigarettes in current or ex-smokers.
Smoking pack-years	Study questionnaire	Number of cigarettes smoked per day divided by 20 and multiplied by the number of years of smoking.
Dependent respiratory variables		
Upper respiratory tract symptoms	Study questionnaire	At least one of the following: rhinorrhoea, nasal congestion, oro-pharyngitis, a hoarse voice, or chronic sneezing in the last two weeks without having the flu.
Ocular symptoms	Study questionnaire	The presence of itchy/watery eyes in the last two weeks.
Lower respiratory tract symptoms		
Shortness of breath	Study questionnaire	Breathlessness walking up a slight hill.
Chronic cough	Study questionnaire	A cough most days for 3 consecutive months or more during the year.
Chronic bronchitis	Study questionnaire	Productive cough for at least three months in a year for at least two consecutive years.
Wheezy chest	Study questionnaire	A wheezing or whistling chest in the last year.
Self-reported diagnosis by a healthcare practitioner		
Emphysema or chronic obstructive pulmonary disease (COPD)	Study questionnaire	Diagnosed or treated by a nurse or a doctor.
Asthma	Study questionnaire	Diagnosed or treated by a nurse or a doctor.
Pulmonary tuberculosis	Study questionnaire	Diagnosed or treated by a nurse or a doctor.
Objective findings of disease		
COPD on spirometry	Spirometry	Forced expiratory volume in one second (FEV1)/ forced vital capacity (FVC) ratio < lower limit of normal using the Global Lung Function Initiative (GLI) 2012 reference equations
Radiological evidence of probable tuberculosis	Chest X-ray	Typical signs of past or present tuberculosis, such as presence of a focal infiltrate, cavity formation, hilar adenopathy, or a miliary pattern. A positive diagnosis required agreement by 2 of the 3 readers.
Radiological silicosis	Chest X-ray	Diagnosis based on the ILO International Classification of Radiographs of Pneumoconioses. Silicosis was defined as a profusion of 1/0 or more reported by at least two of the three readers.

COPD = Chronic Obstructive Pulmonary Disease, FEV1= Forced expiratory volume in one second, FVC=– forced vital capacity, GLI= Global Lung Initiative, ILO = International Labour Organisation.

**Table 2 ijerph-17-02240-t002:** Characteristics and investigations of 310 study participants by exposure group.

Characteristics	Riverlea		Ennerdale
	High Exposure	Medium Exposure	Low Exposure
	*n* (%)	*n* (%)	*n* (%)
Population Size	93 (30.0)	133 (42.9)	84 (27.1)
Median age in years (IQR)	49 (22–72) Range (18–81)	53 (21–81) Range (18–88)	46 (20–68) Range (19–73)
Sex			
Men	36 (38.7)	50 (37.6)	27 (32.1)
Women	57 (61.3)	83 (62.4)	56 (67.9)
Smoking history	60 (64.5)	58 (43.6)	39 (46.4)
Smoking pack-years			
Never smoked (<1 pack-year)	35 (37.6)	77 (57.9)	45 (53.6)
1–10	24 (25.8)	22 (16.5)	22 (26.2)
11–20	15 (16.3)	20 (15.0)	10 (11.9)
>20	19 (20.4)	14 (10.5)	7 (8.3)
Education			
None/Primary	21 (23.1)	14 (10.5)	10 (11.9)
Secondary	65 (71.4)	82 (61.7)	56 (66.7)
Tertiary	5 (5.5)	37 (27.8)	18 (21.4)
Monthly mean income for households			
<R1000	36 (39.6)	27 (20.3)	15 (17.9)
R1000–R5000	40 (43.9)	34 (25.6)	28 (33.3)
R5001–R9999	11 (12.1)	40 (30.10	25 (29.8)
≥R10000	4 (4.4)	32 (24.0)	16 (19.0)
Median years lived in exposure group (IQR)	38 (15–68) Range (2–76)	20 (3–65) Range (1–80)	16 (4–44) Range (2–61)
Reported dustiness outdoors during windy weather	85 (93.4)	119 (90.2)	75 (89.3)
Reported dustiness indoors during windy weather	58 (63.7)	86 (65.2)	55 (65.5)
Use of biomass fuels	3 (4.9)	24 (21.6)	20 (24.4)
House had an asbestos roof	65 (71.4)	39 (29.6)	1 (1.2)
Occupational history of exposure to vapour, gas, dust, or fumes	49 (53.9)	37 (27.8)	15 (17.9)
No. of people who participated in chest X-rays	55 (59.1)	69 (51.9)	53 (63.1)
No. of people who participated in spirometry	43 (46.2)	59 (44.3)	47 (56.0)

IQR = Inter-quartile range.

**Table 3 ijerph-17-02240-t003:** Descriptive and univariate analysis with crude odds ratio and 95% confidence intervals of respiratory symptoms, self-reported respiratory conditions diagnosed by a healthcare practitioner (HCP), and objective findings of respiratory conditions by exposure group.

Variables	Riverlea		Ennerdale	Crude Odds Ratio (95% CI)		
	High Exposure	Medium Exposure	Low Exposure	High vs. Low Exposure	High vs. Medium Exposure	Medium vs. Low Exposure
	*n* (%)	*n* (%)	*n* (%)			
Population Size	93 (30.0)	133 (42.9)	84 (27.1)			
Upper respiratory tract symptoms	63 (67.7)	59 (44.4)	34 (40.5)	**2.83 (1.58–5.08)**	**2.63 (1.52–4.58)**	1.17 (0.67–2.04)
Ocular symptoms	30 (32.3)	31 (23.3)	13 (15.5)	**2.63 (1.52–4.58)**	1.57 (0.87–2.83)	1.66 (0.81–3.39)
Shortness of breath	30 (32.6)	31 (23.5)	23 (27.4)	1.26 (0.66–2.41)	1.57 (0.87–2.83)	0.81 (0.43–1.51)
Chronic cough	32 (34.4)	21 (15.8)	27 (32.1)	1.11 (0.59–2.07)	**2.80 (1.49–5.27)**	0.40 (0.21–0.76)
Chronic bronchitis	19 (20.4)	10 (7.5)	12 (14.3)	1.54 (0.70–3.40)	**3.16 (1.39–7.16)**	0.49 (0.21–1.19)
Wheezy chest	40 (43.5)	28 (21.2)	16 (19.1)	**3.21 (1.62–6.34)**	**2.83 (1.58–5.08)**	1.13 (0.57–2.25)
Self-reported HCP ^a^ diagnosis of emphysema or COPD	8 (8.6)	10 (7.5)	7 (8.3)	1.03 (0.36–2.99)	1.16 (0.44–3.05)	0.89 (0.33–2.45)
Self-reported HCP diagnosis of asthma	8 (8.6)	10 (7.5)	7 (8.3)	1.03 (0.36–2.99)	1.16 (0.44–3.05)	0.89 (0.33–2.45)
Self-reported HCP diagnosis of TB ^b^	4 (4.3)	4 (3.0)	6 (7.1)	0.58 (0.16–2.15)	1.45 (0.35–5.95)	0.40 (0.11–1.47)
	*n* = 55	*n* = 69	*n* = 53			
Radiological evidence of probable TB	6 (10.9)	12 (17.4)	4 (7.5)	1.50 (0.40–5.65)	0.58 (0.20–1.67)	2.58 (0.78–8.51)
Evidence of silicosis on chest X-ray	0 (0)	0 (0)	0 (0)			
	*n* = 43	*n* = 59	*n* = 47			
COPD on spirometry	8 (18.6)	6 (10.2)	5 (10.6)	1.92 (0.58–6.40)	2.00 (0.67–5.95)	0.95 (0.27–3.33)

Median FEV1 (IQR)	98 (59–121) Range (45–132)	91 (48–110) Range (30–121)	93 (75–120) Range (25–143)	-	-	-
Median FVC (IQR)	102 (71–134) Range (57–154)	94 (60–117) Range (48–120)	98 (79–119) Range (22–138)	-	-	-

^a^ HCP = Healthcare practitioner (nurse or doctor); ^b^ TB = Tuberculosis; COPD = Chronic Obstructive Pulmonary Disorder; IQR = Inter Quartile Range; CI = Confidence interval bolded values indicate significant results *p* < 0.05.

**Table 4 ijerph-17-02240-t004:** Multivariate logistic regression models of the associations between exposure group and respiratory health outcomes with adjusted odds ratios and 95% confidence intervals.

Respiratory Health Outcomes	High vs. Low Exposure	High vs. Medium Exposure	Medium vs. Low Exposure
Upper respiratory tract symptoms	**2.76 (1.28–5.97)**	**2.12 (1.01–4.42)**	1.31 (0.70–2.44)
Ocular symptoms	**4.68 (1.87–11.68)**	**3.01 (1.32–6.89)**	1.55 (0.70–3.43)
Shortness of breath	1.52 (0.67–3.44)	2.21 (0.997–4.91)	0.69 (0.34–1.39)
Chronic cough	0.80 (0.34–1.84)	2.18 (0.93–5.15)	**0.36 (0.17–0.77)**
Chronic bronchitis	1.76 (0.59–5.31)	**4.57 (1.35–15.48)**	0.39 (0.13–1.16)
Wheezy chest	**3.78 (1.60–8.96)**	**3.60 (1.60–8.11)**	1.05 (0.48–2.32)
Self-reported diagnosis of COPD or emphysema by HCP ^a^	0.69 (0.17–2.78)	1.14 (0.28–4.71)	1.67 (0.51–5.46)
Self-reported diagnosis of asthma by HCP	0.73 (0.18–2.99)	0.96 (0.25–3.72)	0.77 (0.25–2.37)
Self-reported diagnosis of TB ^b^ by HCP	1.80 (0.33–9.81)	0.52 (0.12–2.18)	3.45 (0.83–14.41)
Radiological evidence of probable TB	1.86 (0.34–10.35)	0.46 (0.11–2.00)	4.03 (0.93–17.49)
COPD on spirometry	**8.17 (1.01–65.85)**	3.40 (0.59–19.62)	2.41 (0.37–15.54)

^a^ HCP = Healthcare practitioner (nurse or doctor); ^b^ TB = Tuberculosis; COPD = Chronic Obstructive Pulmonary Disease; Bolded values indicate significant results *p* < 0.05.

**Table 5 ijerph-17-02240-t005:** Simple and multivariate logistic regression models of the associations between cumulative exposure intensity and respiratory health outcomes.

Respiratory Health Outcomes	Crude OR (95% CI)	aOR * (95% CI)
Upper respiratory tract symptoms	**1.0032 (1.0010–1.0053)**	**1.0034 (1.00051–1.0064)**
Ocular symptoms	**1.0028 (1.00050–1.0051)**	1.0031 (0.99987–1.0063)
Shortness of breath	**1.0029 (1.00064–1.0052)**	1.0027 (0.99958–1.0058)
Chronic cough	1.00088 (0.9986–1.0032)	0.99946 (0.9962–1.0027)
Chronic bronchitis	1.0025 (0.99961–1.0053)	1.0020 (0.9978–1.0062)
Wheezy chest	**1.0039 (1.0017–1.0062)**	**1.0043 (1.0011–1.0075)**
Self-reported diagnosis of COPD or emphysema by HCP ^a^	1.0033 (0.99989–1.0067)	1.0017 (0.9963–1.0070)
Self-reported diagnosis of asthma by HCP	1.0020 (0.9985–1.0056)	0.9987 (0.9934–1.0039)
Self-reported diagnosis of TB ^b^ by HCP	1.0018 (0.9978–1.0059)	1.0005 (0.9949–1.0061))
Radiological evidence of probable TB	1.0019 (0.9978–1.0059)	1.00084 (0.9953–1.0065)
COPD on spirometry	1.0036 (0.99935–1.0078)	**1.010 (1.0014–1.019)**

^a^ HCP = Healthcare practitioner (nurse or doctor); ^b^ TB = Tuberculosis; COPD = Chronic Obstructive Pulmonary Disorder; Bolded values indicate significant results * *p* < 0.05. Note: All models adjusted for age, sex, monthly income, smoking pack-years, use of biomass fuels in the house, and occupational exposure to gases, fumes, and dust. COPD diagnosed on spirometry was additionally adjusted for radiological features of tuberculosis on chest X-ray;.

**Table 6 ijerph-17-02240-t006:** Multiple linear regression model of the FEV1 and FVC percent predicted on the cumulative exposure index.

Variables	β	95% CI	*p*-Value
FEV1 Cumulative Exposure Index (CEI)	0.029	−0.022-0.080	0.27
Interaction term			
smoking pack-years # CEI			
<1	Ref		
1–10	0.00014	-0.080–0.080	0.997
11–20	**−0.19**	**−0.30–(–0.077)**	**0.001**
>20	−0.040	−0.12–0.38	0.31
FVC Cumulative Exposure Index (CEI)	**0.062**	**0.016–0.11**	**0.008**
Interaction term			
smoking pack-years # CEI			
<1	Ref		
1–10	3.39e–06	-0.077-0.077	1.00
11–20	**−0.18**	**−0.27–(–0.093)**	**<0.001**
>20	–0.036	–0.13–0.054	0.43

CEI = Cumulative Exposure Index, # = multiplied; Note: Model adjusted for monthly income, smoking pack-years, use of biomass fuels in the house, and occupational exposure to gases, fumes and dust. FEV1 F < 0.001, R^2^ = 0.31. FVC F < 0.001, R^2^ = 0.31. Bolded values indicate significant result *p*-value < 0.05

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
