# Peer review of "Respiratory Health in a Community Living in Close Proximity to Gold Mine Waste Dumps, Johannesburg, South Africa"

_ijerph, 2020, doi:10.3390/ijerph17072240_

Round 1

Reviewer 1 Report

In their manuscript “Respiratory Health in a Community Living in Close Proximity to Gold Mine Waste Dumps, Johannesburg, South Africa” the authors investigated the association of respiratory health with the exposure to gold mine dump dust in populations living within 500m, 500m to 1.5km or more than15 km from the source of exposure.

The state of the art on the subject is well done and the data collected correctly described. However, significant improvements are needed for : the paper aims, the results presentation and the summary of the major findings (first paragraph of the discussion).

Concerning the paper objectives (last paragraph of the introduction), please be more precise on what aspect the study bring new information (distance of exposure and respiratory health)

In the Results section, the results cited in the tables have to fit those described in the text. The middle exposure group has to be mentioned in the lines 200 to 208.

The tables’ number had to correspond to those cited in the text.

In the discussion, the main findings of the study described in the first paragraph has to correspond to the results.

Additional comments was added directly on the pdf manuscript.

Reviewer 2 Report

The study considered a relatively small population of 310 individuals divided into three groups. The results presented are mostly from a questionnaire, in addition to spirometry and chest radiography data from about 50% of the population. Given the outcome variables, these numbers do not support a strong conclusion. The results are nevertheless novel and, I think, provide a first insight into the topic.

Several points may be improved:

The abstract could be improved by providing several key points.

As the fundamental independent variable is "min dump dust" this should be clearly defined and, preferably, quantified in the locations where the three groups studied resided.

The limited sample size should be mentioned from the onset, and perhaps the study classified as pilot because of the majority of the information was retrieved from a questionnaire.

The final part of the discussion mentions respondents and non-respondents; clarify how the information was available from non-respondents.

Please refer to the attached pdf for specific comments.

I hope that these comments may help improving the ms.

Round 2

Reviewer 1 Report

The authors made the corrections asked.

However, there is still some inconsistency between the number of tables in the table title and in the text. Please don’t change the Table number in the Table title if it doesn’t have to. Thus,  actually Table 2 cited in the text correspond to Table 1, etc.

By the way, please consider to indicate what the values represents in all Tables. Indeed, for Table 2, for example, “N=93 (%)” annotation is ambiguous as it is an information in itself. It corresponds to the population size and no % is added between brackets. Thus, the information has to appear as N=93 (30 %) or N=93. To indicate what the values represents in Tables you can either mention at the bottom of the table that “all the characteristics considered correspond to n (%) unless otherwise stated in the table” or on each line you can add after the characteristic name n(%) (e.g. Men, n(%))

Author Response

To:       Reviewer 1

                                                                                                                                               Date: March 11, 2020

Re: Reviewer 1 Comments

Thank you for your thorough feedback and for picking up all the issues we missed. I am confident that we addressed all your concerns.

Sincerely,

Dr Samantha Iyaloo

Occupational Medicine Specialist

MBChB, MPH, FCPHM (Occupational Medicine), MMed.

National Institute for Occupational Health, National Health Laboratory Service

Tel: +27(0)11 712 6550, 082 584 6846  

samanthai@nioh.ac.za |  www.nioh.ac.za   | www.nhls.ac.za

Practice Number: 5200296

National Institute for Occupational Health

Promoting Healthy, Happy, Safe and Sustainable Workplaces

Medicine and Occupational Hygiene Sections, NIOH

Comments and Suggestions for Authors

The authors made the corrections asked.

However, there is still some inconsistency between the number of tables in the table title and in the text. Please don’t change the Table number in the Table title if it doesn’t have to. Thus,  actually Table 2 cited in the text correspond to Table 1, etc.

Thank you. I apologize for not correcting this in the first round corrections. The track changes confused me. All Tables changed and they correspond to the correct table in the text. This was double checked.

By the way, please consider to indicate what the values represents in all Tables. Indeed, for Table 2, for example, “N=93 (%)” annotation is ambiguous as it is an information in itself. It corresponds to the population size and no % is added between brackets. Thus, the information has to appear as N=93 (30 %) or N=93. To indicate what the values represents in Tables you can either mention at the bottom of the table that “all the characteristics considered correspond to n (%) unless otherwise stated in the table” or on each line you can add after the characteristic name n(%) (e.g. Men, n(%))

Yes, point taken. I tried to address this by rearranging the table structure slightly so I could “population size” as a line item. I hope this avoids the confusion. Please see Tables 2 and 3

n (%)

n (%)

n (%)

Population Size

93 (30.0)

133 (42.9)

84 (27.1)
